evolution, ecology, physiology

geometric framework for nutrition, *Drosophila*, unbalanced diet, nutrient assimilation, limiting nutrient, malnutrition

**Author for correspondence:**
Tadeusz J. Kawecki
e-mail: tadeusz.kawecki@unil.ch

# Experimental evolution of post-ingestive nutritional compensation in response to a nutrient-poor diet

Fanny Cavigliasso[1], Cindy Dupuis[1], Loriane Savary[1], Jorge E. Spangenberg[2] and Tadeusz J. Kawecki[1]

[1]Department of Ecology and Evolution, and [2]Institute of Earth Surface Dynamics, University of Lausanne, Lausanne, Switzerland

FC, 0000-0002-7764-4934; CD, 0000-0002-7850-2890; JES, 0000-0001-8636-6414; TJK, 0000-0002-9244-1991

The geometric framework of nutrition predicts that populations restricted to a single imbalanced diet should evolve post-ingestive nutritional compensation mechanisms bringing the blend of assimilated nutrients closer to physiological optimum. The evolution of such nutritional compensation is thought to be mainly driven by the ratios of major nutrients rather than overall nutritional content of the diet. We report experimental evolution of divergence in post-ingestive nutritional compensation in populations of *Drosophila melanogaster* adapted to diets that contained identical imbalanced nutrient ratios but differed in total nutrient concentration. Larvae from 'Selected' populations maintained for over 200 generations on a nutrient-poor diet with a 1 : 13.5 protein : carbohydrate ratio showed enhanced assimilation of nitrogen from yeasts and reduced assimilation of carbon from sucrose than 'Control' populations evolved on a diet with the same nutrient ratio but fourfold greater nutrient concentration. Compared to the Controls, the Selected larvae also accumulated less triglycerides relative to protein. This implies that the Selected populations evolved a higher assimilation rate of amino acids from the poor imbalanced diet and a lower assimilation of carbohydrates than Controls. Thus, the evolution of nutritional compensation may be driven by changes in total nutrient abundance, even if the ratios of different nutrients remain unchanged.

## 1. Introduction

Most animal populations face periods of food shortage when they have to get by with a diet of limited quantity and/or poor quality, resulting in physiological stress and reduced Darwinian fitness. If sufficiently frequent, such periods of undernutrition are likely to be important factors of natural selection, leading to specific physiological or behavioural adaptations. Potential targets of such natural selection include an increased intake of the poor diet and improved efficiency with which nutrients are extracted from the ingested food and absorbed (assimilated) from the gut. For example, some animals show a plastic response to low nutrient content in their diet by increasing their food consumption [1–5] or their gut size [6,7], presumably to increase the volume of food processed and the surface by which nutrients are absorbed [8].

Increased consumption or larger gut will tend to increase the assimilation of all nutrients. However, to maximize fitness, animals typically require different nutrients in specific quantities or ratios. For example, variation in the protein to carbohydrate ratio (P : C) in the diet affects life-history traits even if the caloric content of the diet remains the same [9–14]. Such results underpin the geometric framework of nutrition, which posits that the ratios of different macronutrients derived from the diet are as important for Darwinian fitness as the total calorie intake [12,14–20]. A nutritionally balanced diet—one with nutrient ratios that

maximize fitness—might be achieved by combining complementary food sources that differ in nutritional composition (pre-ingestive regulation [9,18,21]). However, this option may not be available to some populations, leaving them restricted to an imbalanced diet with suboptimal nutrient ratios [16–18]. In such a case, post-ingestive compensation (also called post-ingestive regulation [18]) would be necessary to approach the optimal ratio of nutrients [18,22]. Post-ingestive compensation can occur in two ways [22]. First, organisms may modulate the efficiency of digestion and absorption of different nutrients, as has been demonstrated in several arthropods [23–26]. Second, the organism may modify the metabolic fate of absorbed nutrients. In particular, excess carbohydrates are often converted to body fat [22,27] but may also be excreted or 'burned off' as 'wastage respiration' [22,28]. All these compensating mechanisms may be costly and/or imperfect; thus, in spite of them, animals confronted with an unbalanced diet may still face a trade-off between the consequence of an excess of some nutrients and a deficit of others.

Consequences of nutrient imbalance are likely to be particularly acute if the diet is not only imbalanced but also generally scarce in nutrients. In this paper, we use experimental evolution to study adaptation to such an imbalanced and nutrient-poor diet. Specifically, we ask if and how the overall reduction of nutrient content affects the evolution of nutrient assimilation if the ratio of different nutrients remains unchanged. On the one hand, it has been suggested that the optimal nutrient ratio, at least for generalist species, is independent of the caloric content of the diet [15–18]. One could thus surmise that a population adapted to a rich imbalanced diet should have evolved compensating mechanisms that would redress this imbalance also when confronted with a poor diet with identical nutrient ratios. If so, natural selection should favour a general increase in the assimilation of all nutrients from the poor diet (enhanced assimilation hypothesis) rather than a shift in post-ingestive nutritional compensation mechanisms. Alternatively, even if the nutrient ratios are the same, selection for regulation and compensation might be affected by the overall caloric content of the diet. In particular, a non-optimal ratio may not be too detrimental if there is a lot of nutrients and become more critical when the diet is generally poor, resulting in a stronger selection for compensation. Furthermore, obtaining a minimum amount of each essential nutrient necessary for survival, development and reproduction may become more important on poor diets than the maintenance of nutrient balance. If so, the overall caloric content of an imbalanced diet should affect the evolution of post-ingestive compensation even if the nutrient ratios are the same. Compared to populations adapted to an imbalanced rich diet, populations evolving on a similarly imbalanced poor diet would thus be predicted to increase selectively their effort to assimilate the limiting nutrients, and possibly reduce their assimilation of nutrients that are less scarce (compensation shift hypothesis).

We addressed these alternative hypotheses using experimental evolution with *Drosophila melanogaster*. We used six 'Selected' populations reared on a very poor larval diet for more than 233 generations, as well as six 'Control' populations maintained in parallel on a standard diet, whereby both diets had the same unbalanced P : C ratio. According to [29], *D. melanogaster* larval growth and survival under laboratory conditions are maximized on diets consisting of about 50–80 g of proteins and 34–90 g of carbohydrates l$^{-1}$ of food medium, with a P : C ratio between 1.5 : 1 and 1 : 2. By comparison, the

poor diet we used to raise the Selected populations only contains 2.4 g of proteins and 32.2 g of carbohydrates l$^{-1}$, resulting in a P : C ratio of about 1 : 13.5. Even allowing that the optimal diet of our populations may differ somewhat from that reported in [29], our poor diet is clearly protein-deficient and highly imbalanced. As a consequence, when raised on the poor diet, non-adapted populations suffer high mortality, take almost twice as long to complete larval development and emerge at half the adult weight compared to flies raised on standard diet [30]. The 'standard' diet, on which the Control populations have been reared, contains fourfold more of all nutrients than the poor diet; while more nutrient-rich, it thus has the same imbalanced P : C ratio. It results in high survival (90–95%) and fast egg-to-adult development time of about 10.5 days (compared to about 10 days on optimal diets [31,32]). Thus, while both Selected and Control populations evolved on similarly imbalanced larval diets, Selected populations also had to cope with strong undernutrition. In the course of experimental evolution, the Selected populations have adapted to the poor diet—they develop faster and survive better on the poor diet than the non-adapted Control populations [30,33]. They also evolved a smaller critical size for metamorphosis initiation, which allows them to complete development with less accumulated biomass [34].

We tested for differences in protein and carbohydrate assimilation from a poor and imbalanced diet between larvae from Control and Selected populations. Our approach was based on heavy stable isotopes of carbon and nitrogen ($^{13}$C and $^{15}$N) [35–37]. We quantified the relative amount of carbon assimilated, within a set time, into the larval body tissues from $^{13}$C-enriched sucrose, and of nitrogen assimilated from $^{15}$N-enriched proteins and other nitrogen-containing molecules provided by dietary yeast. Under the 'enhanced assimilation hypothesis' stated above, the Selected populations should show improved assimilation of both sets of nutrients compared to Controls. By contrast, under the 'compensation shift hypothesis', we expected a higher assimilation rate of nitrogen from yeast in Selected populations than in the Controls, but no difference, or even a reduction, in the assimilation rate of carbon from sucrose. The latter result would imply changes in post-ingestive compensation because the homogenized nature of the diet media precluded pre-ingestive compensation.

Our results show that Selected populations assimilated nitrogenous compounds from yeast better than the Control populations, while the opposite held for assimilation of carbon from sucrose. Consistent with this and the fact that excess sugars are usually converted to body fat [27], we found that, at the end of larval development, Selected populations had accumulated lower fat stores than Controls. Finally, by testing the effects of supplementing the poor diet with 50% more yeast or 50% more sucrose, we verified that only yeast and not sucrose was limiting larval development on the poor imbalanced diet. Our results imply that adaptation to poor larval diet was in part mediated by a shift in post-ingestive nutritional compensation arising from both a higher assimilation efficiency of the limiting nutrient and a lower assimilation rate of a non-limiting nutrient.

## 2. Material and methods

### (a) Origin of fly populations

We used six Control and six Selected populations of *D. melanogaster* derived from a laboratory-adapted base population collected

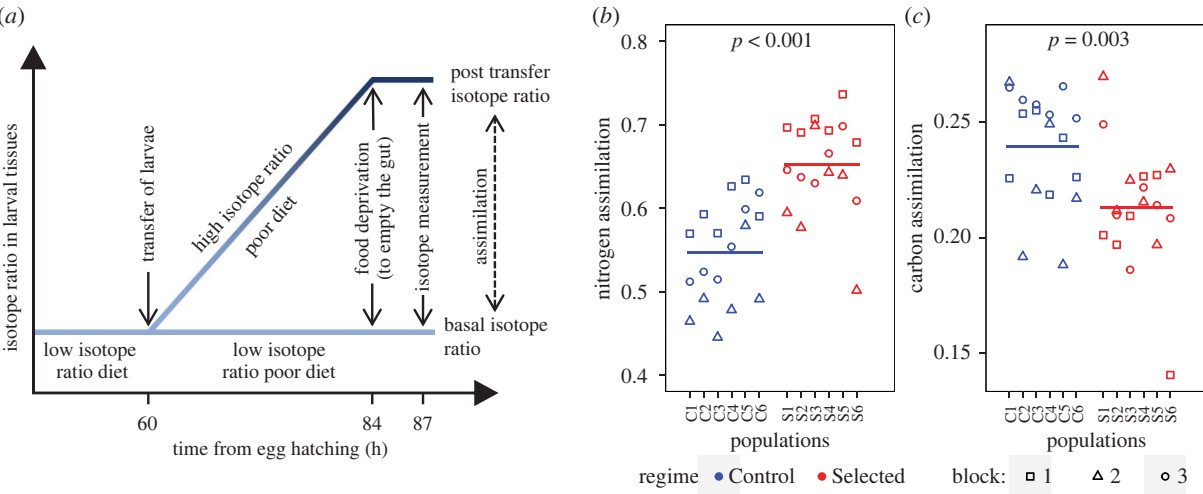

**Figure 1.** Quantifying the assimilation of nitrogen (from yeast) and carbon (from sucrose). (*a*) The principle of the assimilation experiment. After 60 h of growth on standard diet, one group of larvae was transferred to low $\delta^{15}$N|low $\delta^{13}$C poor medium, another to high $\delta^{15}$N|high $\delta^{13}$C poor medium. The thick lines represent hypothetical isotope ratios in the tissues of the two groups of larvae. The difference in isotope ratios relative to the difference in the isotope ratio in the diet reflects the amount of the element assimilated after the transfer. (*b*) Nitrogen and (*c*) carbon assimilation. C1–C6 and S1–S6 refer to the six Control and six Selected populations. Each point corresponds to a single replicate (performed in three blocks indicated by symbol shape). Horizontal lines are the means of Control and Selected populations, respectively. (Online version in colour.)

in Basel, Switzerland, in 1999 and subsequently maintained in the laboratory on the standard diet until the Selected and Control populations were derived in 2005. Details of fly maintenance are described elsewhere [30]. The Control populations have been maintained on a standard diet (15 g agar, 30 g sucrose, 60 g glucose, 12.5 g dry brewer's yeast, 50 g cornmeal, 0.5 g CaCl$_2$, 0.5 g MgSO$_4$, 10 ml Nipagin 10%, 6 ml propionic acid and 20 ml ethanol l$^{-1}$ of water). This corresponds to 9.6 g of protein (including 5.6 g from yeast and 4 g from cornmeal) and 129 g of carbohydrates l$^{-1}$ (contributions from yeast and cornmeal according to suppliers' specifications), resulting in a P : C ratio of 1 : 13.5. The six Selected populations have been maintained on poor larval diet containing one-fourth of the amount of sugars, yeast and cornmeal of the standard diet; thus, the poor diet had the same 1 : 13.5 P : C ratio as the standard diet. After their emergence, adult flies of both selection regimes were transferred to standard diet. Before each experiment, we reared all 12 populations on standard diet for two to four generations (relaxed selection) to avoid maternal effects. To obtain larvae for each experiment, we let 200 adult flies from a given population lay eggs overnight on orange juice-agar plates supplemented with yeast. The desired number of eggs was then transferred onto the experimental media (see below) and inoculated with faeces (OD$_{600}$ = 0.5) from a pool of flies from all populations to ensure homogeneity of larval microbiota [33]. All experiments were performed at 25°C with 50–70% humidity and 12 L : 12 D cycle. The assays reported here were performed after 233–256 generations of the experimental evolution.

## (b) Nutrient assimilation experiment

The design of this experiment is illustrated in figure 1*a* (for detailed protocols, see the electronic supplementary material). We used larval food media containing yeast with different isotope ratios of nitrogen ($^{15}$N/$^{14}$N, expressed as $\delta^{15}$N) and sucrose with different isotope ratios of carbon ($^{13}$C/$^{12}$C, expressed as $\delta^{13}$C). After growing for 2.5 days on a medium with a low isotope ratio for both N and C, some larvae were transferred to a medium with a low isotope ratio (low $\delta^{15}$N|low $\delta^{13}$C medium) and others to a medium with a high isotope ratio (high $\delta^{15}$N|high $\delta^{13}$C medium). Following the transfer, the isotope ratio of the second group of larvae would increase as they fed and assimilated N and C from the high $\delta^{15}$N|high

$\delta^{13}$C medium into their growing tissues. Thus, the difference in $\delta^{15}$N and $\delta^{13}$C between the two groups of larvae would reflect, respectively, the assimilation of nitrogen (mainly in the form of protein) from yeast and carbon from sucrose that occurred since the transfer. Because our interest was to compare the ability to assimilate nutrients from a nutrient-poor diet, the low $\delta^{15}$N| low $\delta^{13}$C and high $\delta^{15}$N|high $\delta^{13}$C media following the transfer of larvae contained a low concentration of yeast and sugar, roughly corresponding to the nutrient content of the poor diet used in the course of experimental evolution. However, to avoid disparities in size and developmental stage, the (low $\delta^{15}$N|low $\delta^{13}$C) medium provided to the larvae prior to the transfer contained a standard amount of nutrients. Before being collected for the isotope analysis, the larvae were deprived of food for 3 h; this ensured that the measured $\delta^{13}$C and $\delta^{15}$N values reflected the composition of larval tissues rather than that of the gut content [38] (electronic supplementary material).

The high $\delta^{15}$N medium was obtained by adding yeast laboratory-grown with $^{15}$N-labelled NH$_3$NO$_2$; the low $\delta^{15}$N medium contained the same amount of yeast laboratory-grown with non-labelled NH$_3$NO$_2$. The difference in $\delta^{13}$C was achieved by using cane sucrose (C$_4$ biomass with relatively high $\delta^{13}$C) versus beet sucrose (C$_3$ biomass with relatively low $\delta^{13}$C) (electronic supplementary material, table S1). In addition to being controlled for isotope content, the media used in this experiment differed from those used in the experimental evolution in that they had a lower concentration of agar and did not contain cornmeal (for the composition of media, see the electronic supplementary material). This was to facilitate rapid extraction of the larvae and their transfer to a new medium necessary for this experiment. Otherwise, the media contained similar amounts of yeast and sugar, at a similarly imbalanced P : C ratio, as the standard and poor diets used in the experimental evolution.

The stable isotope content of larvae and diet was quantified using an established protocol [39,40]; for details, see the electronic supplementary material. Following O'Brien *et al.* [37], we quantified nitrogen and carbon assimilation as the difference in $\delta^{15}$N and $\delta^{13}$C between the larvae transferred to the high $\delta^{15}$N|high $\delta^{13}$C versus low $\delta^{15}$N|low $\delta^{13}$C medium divided by the difference in $\delta^{15}$N and $\delta^{13}$C between high $\delta^{15}$N|high $\delta^{13}$C and low $\delta^{15}$N|low $\delta^{13}$C media (measured separately for each block). This quantity estimates the percentage of nitrogen and of carbon in larval tissues assimilated since the transfer.

### (c) Triglyceride content quantification

The aim of this assay was to compare the amount of carbohydrates stored as triglycerides (TAG) at the end of larval development (at the prepupal stage). Larvae were raised on the poor diet as described above (starting with approximately 200 eggs per bottle with 40 ml of food). We pooled prepupae from four replicate bottles and collected three samples of five prepupae (white puparia with everted spiracles) per population (for Selected population S4 and Control population C4 only two samples were obtained). For one sample of three Control populations (C3, C5 and C6), we could only collect four prepupae. Samples were flash-frozen in liquid nitrogen and stored at −80°C until triglyceride assays. Triglycerides were quantified using a protocol adapted from [41] (electronic supplementary material). Triglyceride content was normalized by dividing by the protein content of the sample (a proxy of body size).

### (d) Limiting nutrient experiment

The aim of this experiment was to compare the degree to which the two nutrient sources (i.e. sugar or yeast) were limiting for larval survival and development, and whether this differed between the Selected and Control populations. To this end, we raised larvae on four diets: poor diet (the same as used during the experimental evolution), poor diet with 1.5-fold more yeast (yeast+), poor diet with 1.5-fold more glucose and sucrose (sugar+), and poor diet with 1.5-fold more of both sugar and yeast (yeast+|sugar+). The comparison of performance between larvae reared on poor diet and those reared yeast+ or sugar+ would tell us which of yeast or sugar is the most limiting nutrient. Furthermore, the inclusion of yeast+|sugar+ would tell us if there is a synergic or antagonistic effect of supplementing the two nutrients.

The experiment was set up with precisely 200 eggs per bottle as described above (two replicate bottles per population and diet). The populations were split among three blocks (two Selected and two Control populations per block) and between two experimenters (each experimenter sorted eggs of one Control and one Selected population per block). We scored emerging adult males and females every day; these data were used to calculate for each replicate the egg-to-adult survival probability, the adult sex ratio and the average sex-specific developmental rate. This last parameter was calculated as the inverse of egg-to-adult development time.

For each bottle of each population and diet, adults were collected within 48 h (or 72 h) at their emergence peak (i.e. the second or third day after the first emergence for most replicates) and we pooled and froze them at −20°C. Then, we randomly picked 20 females and 20 males, dried them at 70°C for 24 h and weighed them as a group on a precision balance (Mettler Toledo, MT5, resolution of 1 μg). For some populations, fewer than 20 males or females were available, and the individual body weight was evaluated by dividing the weight of a group by the number of individuals in the group.

### (e) Statistical analysis

Nitrogen and carbon assimilation estimates, normalized triglyceride content (log-transformed because it was a ratio), developmental rate and adult weight were analysed by fitting linear mixed models (LMM) with Type 3 $F$-tests, using $lmer$ package of R [42]. Survival (number of dead versus alive) and sex ratio (number of males versus females) were analysed with a generalized linear model with a binomial error distribution and likelihood ratio tests, using the $afex$ package [43]. In all analyses, replicate populations were a random factor. Where applicable, sex or diet treatment was included as fixed factors. Experimental block and experimenter, while in principle random, were also modelled as a fixed effect because of small number levels (2 or 3). When applicable, pairwise comparisons were performed with $emmeans$ and $pairs$ functions in R [44]. For details of statistical analysis, see the electronic supplementary material.

## 3. Results

### (a) Nitrogen and carbon assimilation

Based on the isotope ratio data, the Selected larvae showed a 19% higher nitrogen assimilation than Controls ($0.652 \pm 0.014$ versus $0.547 \pm 0.014$, mean ± s.e.; $F_{1,12} = 32.8$, $p < 0.001$; figure 1b). This implies that they extracted nutrients from proteins and other nitrogenous compounds at a faster rate. The levels of $\delta^{15}N$ in larvae of both selection regimes transferred to the high $\delta^{15}N$|high $\delta^{13}C$ medium were closer to the $\delta^{15}N$ of that medium than to $\delta^{15}N$ of the low $\delta^{15}N$|low $\delta^{13}C$ medium (electronic supplementary material, figure S2a and table S1). This indicates that more than 50% of the nitrogen in their bodies was assimilated after the transfer to the high $\delta^{15}N$|high $\delta^{13}C$ medium, as reflected in the nitrogen assimilation values in figure 1b. Basal larval level of $\delta^{15}N$ (i.e. the level in larvae transferred to low $\delta^{15}N$|low $\delta^{13}C$ diet) was indistinguishable between Control and Selected populations ($1.34 \pm 0.18$ versus $1.98 \pm 0.84$; $F_{1,36} = 0.6$, $p = 0.44$; electronic supplementary material, figure S2a).

The Control larvae showed a 12% higher $^{13}C$ assimilation compared to the Selected larvae ($0.239 \pm 0.003$ versus $0.213 \pm 0.007$; $F_{1,36} = 10.9$, $p = 0.002$; figure 1c). These values reflect carbon assimilation from sucrose, as only sucrose differed in $\delta^{13}C$ between the high $\delta^{15}N$|high $\delta^{13}C$ and low $\delta^{15}N$|low $\delta^{13}C$ media. Carbon was obviously also acquired from yeast, which is the main reason why the $\delta^{13}C$ of larvae from the low $\delta^{15}N$|low $\delta^{13}C$ medium (electronic supplementary material, figure S2b) was intermediate between $\delta^{13}C$ of beet sucrose ($\delta^{13}C = -26.5$) and of yeast ($\delta^{13}C = -23.2$; electronic supplementary material, table S1) and closer to the latter. Interestingly, this 'basal' level of $\delta^{13}C$ was higher in Selected than in Control populations ($-23.32 \pm 0.06$ versus $-23.56 \pm 0.10$; electronic supplementary material, figure S2; LMM, $F_{1,12} = 5.2$, $p = 0.041$). This implies that the Selected larvae obtained a higher proportion of their body carbon from yeast than Controls.

### (b) Accumulation of triglycerides

Excess of assimilated carbon is converted into triglycerides, and thus, the reduced assimilation of carbohydrates from the poor diet by the Selected larvae should be reflected in a lower triglyceride accumulation. To test this prediction, we quantified the amount of triglycerides after completion of the larval development on the poor diet (at the prepupal stage). Selected prepupae showed a 45% lower triglyceride content (normalized to protein content) than prepupae from Control populations ($0.099 \pm 0.017$ versus $0.180 \pm 0.017$; $F_{1,12.8} = 9.3$, $p = 0.009$; figure 2).

### (c) Limiting nutrient for larval fitness traits

Compared to other populations, populations C4, S3 and S4 showed an abnormal delay in development and/or a very low number of emerging flies in this experiment, even though these populations showed qualitatively similar responses to diet as the others (electronic supplementary

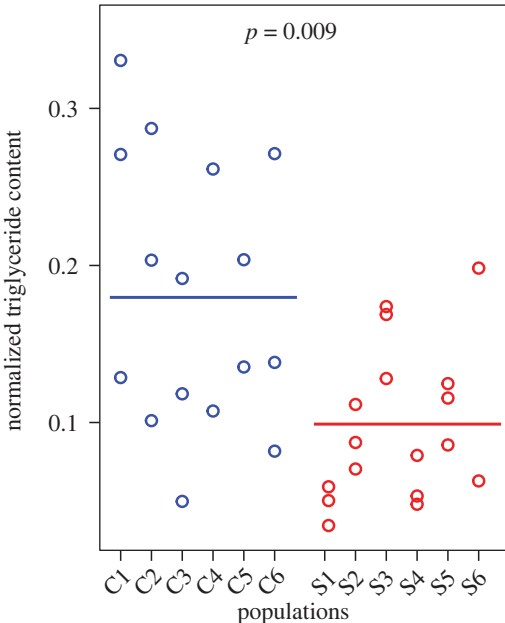

**Figure 2.** Triglyceride (TAG) content (normalized to protein) at the prepupal stage after development on the poor diet. (Online version in colour.)

material, figure S3). These three populations, along with the C3 population, formed one of three blocks of the experiment. Because of this apparent technical issue, we removed this block from our analysis (thus, the analysis is based on four Selected and four Control populations).

As reflected in regime × supplemental yeast interaction ($\chi^2 = 9.0$, $p = 0.003$), supplemental yeast added to the poor diet improved the egg-to-adult survival probability of Control populations by about 14% (figure 3a; electronic supplementary material, table S4; $z = -5.2$, $p < 0.001$) but did not detectably affect the survival of the Selected populations (figure 3a; $z = -0.6$, $p = 0.53$). No effects of sugar supplement were detected; if anything, it tended to worsen egg-to-adult survival probability (figure 3a; electronic supplementary material, table S4; $\chi^2 = 3.0$, $p = 0.082$, 95% confidence interval (CI) of the effect on the logit scale (–0.31, 0.01)). The adult sex ratio did not vary with diets nor selection regime (electronic supplementary material, table S4; all $p > 0.1$), implying that their effects on survival on the two sexes were similar (figure 3b).

As expected based on previous results, the Selected populations showed faster development and smaller adult weight than the Controls, irrespective of the diet and sex (figure 3c,d; $p = 0.002$ and $p < 0.001$, respectively, electronic supplementary material, table S4). In spite of supplemental yeast × sex interaction ($F_{1,44} = 6.6$, $p = 0.014$), supplementing 50% more yeast resulted in faster development of both sexes (figure 3c; $F_{1,7} = 140.4$, $p < 0.001$ for females; $F_{1,7} = 194.1$, $p < 0.001$ for males), although the magnitude of the effect differed between the sexes (supplemental yeast × sex interaction $F_{1,44} = 6.6$, $p = 0.014$). By contrast, the sugar supplement had no detectable effect on the developmental rate (figure 3c; $F_{1,6} = 1.2$, $p = 0.31$, 95% CI on the effect size (–0.003,0.001) for females; $F_{1,6} = 1.7$, $p = 0.24$, 95% CI (–0.002 0.001) for males).

The yeast supplement increased female and male adult dry weight by more than 20%, irrespective of the amount of sugar in the diet (figure 3d; electronic supplementary material, table S4; $p < 0.001$). Although supplemental sugar did not affect male weight (figure 3d; $t = -1.96$ or 0.86, $p =$

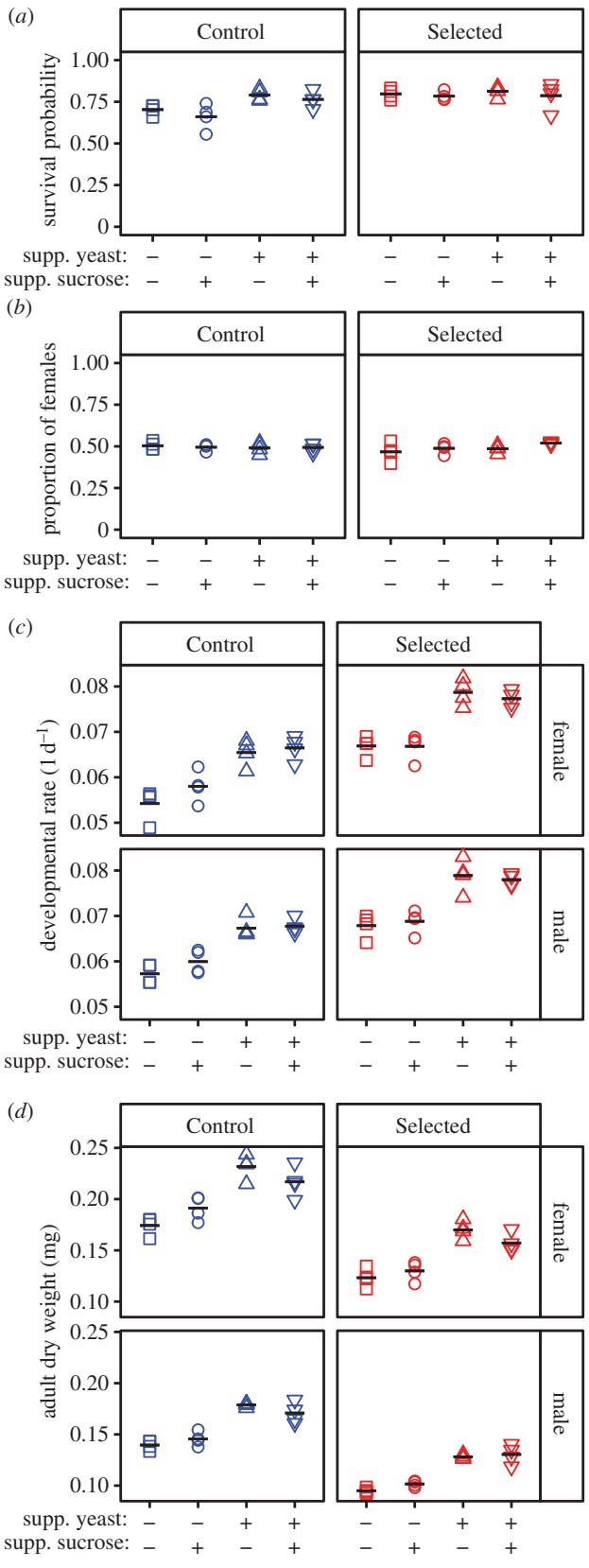

**Figure 3.** Effect of supplementing the poor diet with 50% additional yeast and/or sucrose on the developmental traits of Control and Selected populations. (a) Egg-to-adult survival probability. (b) Proportion of emerging adult females. (c) Sex-specific developmental rate calculated as the inverse of the egg-to-adult development time. (d) Sex-specific adult dry weight. The '+' and '–' signs indicate the presence/absence of a particular nutrient supplement. Each point represents the average per replicate bottle, symbol shape and colour indicate the population and diet, respectively (only four out of six populations per selection regime used, see 'Results'); lines are treatment means. (Online version in colour.)

0.071 or 0.40 when sugar is added alone or together with yeast, respectively), it had an opposite effect for female weight depending on the amount of yeast. The addition of sugar alone in poor diet increased the female dry weight by 8% (figure 3d; $t = 2.9$, $p = 0.012$), while it reduced female dry weight by 7% when there was already 50% more yeast in the diet (figure 3d; $t = 3.4$, $p = 0.005$).

Thus, except for the egg-to-adult survival in the Selected populations, all aspects of performance on poor diet supplemented with 50% more yeast were superior to performance on poor diet supplemented with 50% more sugar (electronic supplementary material, table S5).

## 4. Discussion

The principal aim of this study was to test (i) whether evolutionary adaptation to a poor imbalanced diet would involve an improved ability to assimilate nutrients from the poor diet, and (ii) whether changes in nutrient assimilation would be similar for proteins and carbohydrates, or rather favour assimilation of protein, which is scarcer than carbohydrate in the diet relative to the physiological requirements of the larvae.

We found that larvae from the poor diet-adapted Selected populations assimilated dietary nitrogen faster than larvae from the Control populations, which evolved on a diet with a higher nutrient content but the same imbalanced protein to carbohydrate (P : C) ratio. Nearly all dietary nitrogen is contained in proteins and RNA, which in yeast occur at about a 5 : 1 ratio [45]; proteins and RNA consist of 16% and 14% of nitrogen, respectively [46]. Thus, proteins accounted for about 85% of dietary nitrogen. Although a better assimilation of nitrogen from RNA may have contributed, at least a part of the 19% higher nitrogen assimilation by the Selected larvae compared to Controls must derive from a higher rate of amino acid assimilation.

By contrast, carbon from sucrose was assimilated at a lower rate by Selected than Control populations. This implies that the evolution of poor imbalanced diet did not favour mechanisms that improved total nutrient extraction, but specifically enhanced assimilation of proteins (and possibly other nitrogenous compounds) while reducing assimilation of carbohydrates. These contrasting differences in protein versus carbohydrate assimilation rates between Selected and Control populations support the notion that the Selected populations evolved a greater degree of nutritional compensation at the stage of nutrient assimilation.

Two other results corroborate a shift towards a higher P : C ratio of assimilated nutrients in the Selected populations. First, the 'basal' value of $\delta^{13}C$ was higher in Selected than Control larvae. Because $\delta^{13}C$ was higher in dietary yeast than in beet sucrose (electronic supplementary material, table S1), this implies that the Selected larvae obtained a greater proportion of their carbon from yeast (i.e. a lower proportion from sucrose) than the Controls. Second, at the end of their development on the poor diet, larvae of the Selected populations accumulated about 46% less triglycerides relative to protein than the Controls. Conversion into triglycerides is the default fate of excess of assimilated carbohydrates [27], and the amount of triglycerides usually increases with the decreasing P : C ratio in the diet [47–49]. The first of these results could still in principle be explained by a greater accumulation of carbohydrates from

yeast relative to sucrose; however, only 20% of yeast nutritional carbon is contained in carbohydrates compared to 70% in proteins. The second result might in theory be explained by the carbohydrate-derived carbon being stored by Selected larvae in other compounds, such as glycogen, mono- and disaccharides, or lipids other than triglycerides. While our data do not allow us to exclude these alternative explanations, the greater assimilation of carbon from yeast and the lower adiposity of Selected larvae are most parsimoniously explained as a consequence of the evolving a greater assimilation of amino acids relative to carbohydrates, consistent with the results of stable isotope accumulation.

What could be the physiological mechanisms of this shift in nutritional compensation? In a heterogeneous nutritional environment, D. melanogaster larvae are able to achieve nutritional balance by feeding selectively [29,50]. However, the stable isotope experiment to quantify nutrient assimilation was performed using a homogeneous yeast-sucrose-agar paste, intended to give the larvae no opportunity for selective feeding. Drosophila larvae can also adjust their food intake according to the nutrient content of the diet [4]. However, even if Selected larvae ingested more food than Controls during the assimilation assay reported here, this could not explain the opposite effect on assimilation of proteins versus carbohydrates. It is thus more likely that the evolutionary shift in the assimilation rates is mediated by post- rather than pre-ingestive dietary compensation. One potential mechanism would be a differential investment in digestion of protein versus carbohydrates [23,25]. Several digestive proteases are differentially expressed between Selected and Control larvae (although the direction of the difference is not consistent across enzymes), and protease activity is higher in the guts of Selected and Control larvae, although this is only observed in a germ-free state [33]. By contrast, amylase activity in the gut of Selected larvae is lower than in Controls [33]. Changes in assimilation of nutrients might also be mediated by differential ability to absorb and transport the products across the gut wall and into the haemolymph. Finally, the difference may come from differential allocation and processing of amino acids and monosaccharides once absorbed—rather than contributing to the growing biomass of the body, nutrients can be catabolized to produce energy or even excreted if in excess. This last possibility seems less likely. For example, Selected larvae might have catabolized carbohydrates to a greater extent than Controls to generate more ATP. However, if so, they would have more energy available to spend, e.g. for locomotion; yet, the Selected larvae show reduced locomotor activity [51,52], which suggests that they are more 'thrifty' with expending their energy. One might also imagine that the Control larvae might have generated a greater fraction of their energy from amino acid catabolism while saving the carbohydrates. This explanation seems even less probable as the Control larvae, in particular, suffer from protein deficiency on the poor diet. It thus seems more likely that the shift in protein versus carbohydrate assimilation is mediated by processes involved in nutrient acquisition rather than form differential catabolism of protein versus carbohydrates.

The prediction that imbalanced but homogeneous diets should favour post-ingestive compensation [18,22] has been mostly tested experimentally by studying phenotypically plastic responses to diets with different P : C ratios. Several such studies showed plastic responses in a direction consistent with post-ingestive compensation [3,23,25,26,47,49],

suggesting that they were adaptive without demonstrating it directly. To our knowledge, only one study has demonstrated the experimental evolution of post-ingestive compensation in response to imbalanced diet. While non-adapted *Plutella* caterpillars raised on diet with a low P : C accumulate excess body fat, this tendency became strongly reduced in populations allowed to evolve on the imbalanced diet for multiple generations; the opposite was observed in populations evolved on a high P : C ratio [53]. This is a direct demonstration that an imbalanced diet favours the evolution of compensatory physiological mechanisms.

In the present study, we also demonstrate the experimental evolution of post-ingestive compensation, which supports its adaptive nature; however, in our case, it evolved differently between diets with different caloric content but an identical P : C ratio. Given the population sizes and the number of generations, the response to selection must have been essentially entirely based on genetic variants already segregating in the base population [54], and thus also present, at least initially, in the Control populations. This raises the question of why did the Control populations not evolve a similar degree of nutritional compensation—i.e. a higher assimilation of the scarcer nutrient, the proteins—than those maintained on the standard diet, despite the same P : C ratio. First, the optimal P : C ratio in the assimilated nutrients might be the same under the two conditions, but the fitness consequences of deviating from it might be smaller—and thus selection on the compensating mechanisms weaker—if the total caloric content is sufficient. The resulting selection might be too weak to compensate for putative costs of compensating mechanisms. Second, the optimal P : C ratio might actually be different on very nutrient-poor diets because such diets may favour different life-history strategies than diets whose total nutrient content is less marginal. In particular, the Selected populations evolved faster development at the expense of a smaller adult body size and reduced fecundity [30,34], a trade-off that may have prioritized fast assimilation of protein over laying down lipid stores. This trade-off might have been less acute in the Control populations, which face no pressure to develop rapidly [30].

Finally, it is possible that macronutrient balance becomes less important on diets whose protein content is marginal. Our results demonstrate that protein and other yeast-derived nutrients in the poor diet are limiting—supplementing the poor diet with 50% more yeast improves larval development and, for the Control populations, survival. By contrast, supplementing the poor diet with sucrose had no effect on the performance of the poor diet-adapted Selected larvae (although it tended to improve the growth of Control larvae). This suggests that, rather than optimizing the P : C ratio of the assimilated nutrients, the selection on those populations may have acted to maximize the amount of assimilated protein. This is in accordance with the finding that *Drosophila* larvae regulate the amount of protein intake more tightly than the carbohydrates [4,10,29]. In such a scenario where selection acted to improve protein assimilation irrespective of carbohydrate levels, the evolution of reduced carbohydrate assimilation would not have been driven by the harmful effects of excess carbohydrates, but by costs of carbohydrate digestion and absorption from the gut. It should be noted that much (if not most) of carbohydrate digestion by *Drosophila* larvae occurs externally, by salivary gland enzymes secreted by the larvae onto the food [55–57]. Because larvae form feeding aggregations, these secreted enzymes—themselves proteins—are public goods that could be exploited by 'cheater' larvae that do not secrete them [58]. The extreme scarcity of protein in the poor diet might thus have favoured reduced investment of scarce amino acids in these public goods.

While the questions about the specific factors driving the experimental evolution of changes in nutrient assimilation in our study and the underlying physiological mechanisms remain unresolved, our results allow two firm conclusions. First, exposure to a poor and imbalanced diet over generations can lead to the evolution of improved ability to extract and assimilate scarce nutrients, even in larvae of holometabolous insects, which are often thought to be optimized for converting food into their own biomass. Second, adaptation to an imbalanced diet can involve evolutionary changes in post-ingestive nutritional compensation; however, the ratio of nutrients in the diet is not the only factor driving these changes.

Data accessibility. Data are available from the Dryad Digital Repository: https://dx.doi.org/10.5061/dryad.8cz8w9gnq [59].

Authors' contributions. F.C., C.D. and T.J.K. designed the study and analysed the data; F.C., C.D. and L.S. carried out the experiments; J.E.S. performed stable isotope measurements; F.C., C.D. and T.J.K. wrote the paper; all authors approved the submitted manuscript.

Competing interests. We declare we have no competing interests.

Funding. This work has been supported by the Swiss National Science Foundation grants nos. 31003A_162732 and 310030_184791.

Acknowledgements. The authors thank B. Erkosar for advice, J. Cergneux for help with experiments and four anonymous reviewers for their comments on an earlier version of the manuscript.

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
