## [Reviewer comments · Proceedings of the Royal Society B: Biological Sciences]

Review History

RSPB-2020-2158.R0 (Original submission)

Review form: Reviewer 1

Recommendation

Major revision is needed (please make suggestions in comments)

Scientific importance: Is the manuscript an original and important contribution to its field?

Excellent

General interest: Is the paper of sufficient general interest?

Excellent

Quality of the paper: Is the overall quality of the paper suitable?

Excellent

Is the length of the paper justified?

Yes

Should the paper be seen by a specialist statistical reviewer?

No

Do you have any concerns about statistical analyses in this paper? If so, please specify them explicitly in your report.

No

It is a condition of publication that authors make their supporting data, code and materials available - either as supplementary material or hosted in an external repository. Please rate, if applicable, the supporting data on the following criteria.

Is it accessible?

Yes

Is it clear?

Yes

Is it adequate?

Yes

Do you have any ethical concerns with this paper?

No

Comments to the Author

This manuscript makes use of a long-term selection experiment in fruit flies to understand how nutrient assimilation changes with adaptation to poor diet. The authors show that the selected flies increase their nitrogen assimilation and decrease their carbohydrate assimilation, and further that when reared on poor diet the selected lines stores less triglycerides. Finally, they demonstrate that yeast, the primary protein source, is limiting for most larval traits in both the control and selected lines reared on poor diets. I think this manuscript presents novel and important findings that will be relevant to many in the field, however I have some suggestions that I feel should be addressed before publication.

The arguments around the optimal conditions for larval traits are not based on *Drosophila* populations used for these experiments. Recent studies have shown that populations differ in the way they respond to quantitative variation in nutrition. This makes it difficult to say for certain that for your lab population the diets are imbalanced or off target. I think this needs to be acknowledged.

Lines 124-126: I'm not sure I understand the selected diet. Here you state that the Selected lines were maintained on a diet that contained $\frac{1}{4}$ of the yeast, sugar, and cornmeal from the control diet. Because you've reduced all ingredients by the same amount you haven't actually changed the P:C ratio from the control diet right? So you are comparing between a rich imbalanced and poor imbalanced diet? Given that most nutritional geometry experiments compare across P:C ratios (for the same caloric value), I think you need to make this very clear in the introduction that you are looking at selection to a total macronutrient dilution. Because larval traits are driven to a large extent by the protein concentration (and not the interactions between protein and carbohydrate) in their diets, I think this is still reasonable. However, you could also frame this as larvae adapt to low calorie diets by increasing nitrogen assimilation and decreasing carbohydrate assimilation. In fact, this argument would speak directly to one of the central themes in nutritional geometry: that many of the effects attributed to caloric dilution actually are a product of macronutrient concentration and have nothing to do with calories.

Lines 241-243: Why were the lines not compared on the rich diet as well? Would you expect them to differ? I think it's important to include in this portion of the results that triglyceride levels were measured in prepupae that had been raised on the poor diet. I realise that this is stated in the methods, but the reader doesn't always remember these details.

Lines 255-257: The effects of sugar supplementation aren't significant.

In the discussion, I think the points about the effects of P:C ratios are hard to make, because these weren't varied in your experimental design (the poor and control diet are on the same nutrient rail). I think these parts should be toned down, and proposed as suggestions.

Minor comments:

Lines 44-45, 61, 364-365: The geometric framework for nutrition isn't really a theory. Rather, it is a framework that allows you to describe how traits respond to quantitative variation in nutrients. This framework has led to several hypotheses, like the protein leveraging hypothesis, but doesn't postulate any particular response to diet per se. Initially, this framework was devised to understand how animals make foraging decisions. These studies found that animals often make foraging decisions that are based around achieve an intake target of one or a combination of nutrients. More recently it has been adopted to understand how life history traits respond to quantitative variation in nutrient conditions. It's important to note that, while uncommon, intake targets can theoretically revolve around maintaining constant calorie concentrations.

Review form: Reviewer 2

Recommendation

Accept with minor revision (please list in comments)

Scientific importance: Is the manuscript an original and important contribution to its field?

Excellent

General interest: Is the paper of sufficient general interest?

Good

Quality of the paper: Is the overall quality of the paper suitable?

Good

Is the length of the paper justified?

Yes

Should the paper be seen by a specialist statistical reviewer?

No

Do you have any concerns about statistical analyses in this paper? If so, please specify them explicitly in your report.

No

It is a condition of publication that authors make their supporting data, code and materials available - either as supplementary material or hosted in an external repository. Please rate, if applicable, the supporting data on the following criteria.

Is it accessible?

Yes

Is it clear?

Yes

Is it adequate?

Yes

Do you have any ethical concerns with this paper?

No

Comments to the Author

In this manuscript, the authors describe an experimental evolution study in the fly *Drosophila melanogaster* to test for the evolution of post-ingestive nutritional compensation as an adaptation to lowered total dietary nutrient content at the larval stage.

After between 233 and 256 generations on either ancestral ($n=6$), or low-nutrient diet ($n=6$), with imbalanced P:C ratio of 1:13 for all diets, flies evolved enhanced assimilation of nitrogen from dietary yeast and reduced assimilation of carbon from sucrose. Responses were measured using analyses of stable nitrogen and carbon isotopes, triglyceride levels, and survival and development rate.

I would have wished to read about the possible mechanisms of the observed evolutionary changes. If this is based mainly on standing genetic variation that existed in the stock lab population at the start of the experiment, is there a reason why a more efficient uptake of nitrogen wouldn't evolve, even under much lower selection, i.e. is there an assumed trade-off?

Minor comments:

Line 90: Referring to the control diet' nutrient concentration, "thus, it has the same imbalanced P:C ratio": I don't see how 'thus' makes sense here, linking to the previous statement about nutrient concentration. Also, the description of the control diet feels a bit misplaced here. It would be better to fit it in earlier, before mentioning the evolved traits (lines 86-88).

Line 123: Add P:C ratio the given values equate to (1:13.5)

Line 131: Delete "them".

Lines 209-217: It would very helpful and good practice to publish the R code for all analyses. Why was triglyceride level and development rate not modelled in glmers, too, since they're ratios, as is survival? Were glmer with binomial error checked for overdispersion? Were any assumptions checked, and if so, how?

Line 336: "thrifty" needs more explanation.

Line 339: Typo: 'Is'.

Figure2 1b,c, 2: Add SE or 95% CI.

Table S3: Check consistency of interaction symbol (: or x).

Decision letter (RSPB-2020-2158.R0)

07-Oct-2020

Dear Dr Kawecki:

I am writing to inform you that your manuscript RSPB-2020-2158 entitled "Experimental evolution of post-ingestive nutritional compensation in response to a nutrient-poor diet" has, in its current form, been rejected for publication in Proceedings B.

This action has been taken on the advice of referees, who have recommended that substantial revisions are necessary. With this in mind we would be happy to consider a resubmission, provided the comments of the referees are fully addressed. However please note that this is not a provisional acceptance.

Sincerely,
 Professor Hans Heesterbeek
 mailto: proceedingsb@royalsociety.org

Associate Editor
 Board Member: 1
 Comments to Author:

I think this is an interesting study that has a potential to contribute towards literature in the field of evolutionary responses to nutrition.

Two experts provided what I see as generally positive evaluation of this work. However, they also made suggestions towards improvement and expressed some concerns that need to be addressed. I suggest that the authors should pay close attention to the comments. I am particularly interested in their response to comment by Reviewer 1 regarding population differences in what constitutes optimal diet and how this may affect the interpretation of the data. Furthermore, the same reviewer makes important arguments regarding the interpretation of the results in the light of what we learned so far from nutritional geometry studies in *Drosophila*.

Reviewer(s)' Comments to Author:
 Referee: 1

Comments to the Author(s)
 This manuscript makes use of a long-term selection experiment in fruit flies to understand how nutrient assimilation changes with adaptation to poor diet. The authors show that the selected

flies increase their nitrogen assimilation and decrease their carbohydrate assimilation, and further that when reared on poor diet the selected lines stores less triglycerides. Finally, they demonstrate that yeast, the primary protein source, is limiting for most larval traits in both the control and selected lines reared on poor diets. I think this manuscript presents novel and important findings that will be relevant to many in the field, however I have some suggestions that I feel should be addressed before publication.

The arguments around the optimal conditions for larval traits are not based on *Drosophila* populations used for these experiments. Recent studies have shown that populations differ in the way they respond to quantitative variation in nutrition. This makes it difficult to say for certain that for your lab population the diets are imbalanced or off target. I think this needs to be acknowledged.

Lines 124-126: I'm not sure I understand the selected diet. Here you state that the Selected lines were maintained on a diet that contained $\frac{1}{4}$ of the yeast, sugar, and cornmeal from the control diet. Because you've reduced all ingredients by the same amount you haven't actually changed the P:C ratio from the control diet right? So you are comparing between a rich imbalanced and poor imbalanced diet? Given that most nutritional geometry experiments compare across P:C ratios (for the same caloric value), I think you need to make this very clear in the introduction that you are looking at selection to a total macronutrient dilution. Because larval traits are driven to a large extent by the protein concentration (and not the interactions between protein and carbohydrate) in their diets, I think this is still reasonable. However, you could also frame this as larvae adapt to low calorie diets by increasing nitrogen assimilation and decreasing carbohydrate assimilation. In fact, this argument would speak directly to one of the central themes in nutritional geometry: that many of the effects attributed to caloric dilution actually are a product of macronutrient concentration and have nothing to do with calories.

Lines 241-243: Why were the lines not compared on the rich diet as well? Would you expect them to differ? I think it's important to include in this portion of the results that triglyceride levels were measured in prepupae that had be raised on the poor diet. I realise that this is stated in the methods, but the reader doesn't always remember these details.

Lines 255-257: The effects of sugar supplementation aren't significant.

In the discussion, I think the points about the effects of P:C ratios are hard to make, because these weren't varied in your experimental design (the poor and control diet are on the same nutrient rail). I think these parts should be toned down, and proposed as suggestions.

Minor comments:

Lines 44-45, 61, 364-365: The geometric framework for nutrition isn't really a theory. Rather, it is a framework that allows you to describe how traits respond to quantitative variation in nutrients. This framework has led to several hypotheses, like the protein leveraging hypothesis, but doesn't postulate any particular response to diet per se. Initially, this framework was devised to understand how animals make foraging decisions. These studies found that animals often make foraging decisions that are based around achieve an intake target of one or a combination of nutrients. More recently it has been adopted to understand how life history traits respond to quantitative variation in nutrient conditions. It's important to note that, while uncommon, intake targets can theoretically revolve around maintaining constant calorie concentrations.

Referee: 2

Comments to the Author(s)

In this manuscript, the authors describe an experimental evolution study in the fly *Drosophila melanogaster* to test for the evolution of post-ingestive nutritional compensation as an adaptation to lowered total dietary nutrient content at the larval stage.

After between 233 and 256 generations on either ancestral (n=6), or low-nutrient diet (n=6), with imbalanced P:C ratio of 1:13 for all diets, flies evolved enhanced assimilation of nitrogen from dietary yeast and reduced assimilation of carbon from sucrose. Responses were measured using analyses of stable nitrogen and carbon isotopes, triglyceride levels, and survival and development rate.

I would have wished to read about the possible mechanisms of the observed evolutionary changes. If this is based mainly on standing genetic variation that existed in the stock lab population at the start of the experiment, is there a reason why a more efficient uptake of nitrogen wouldn't evolve, even under much lower selection, i.e. is there an assumed trade-off?

Minor comments:

Line 90: Referring to the control diet' nutrient concentration, "thus, it has the same imbalanced P:C ratio": I don't see how 'thus' makes sense here, linking to the previous statement about nutrient concentration. Also, the description of the control diet feels a bit misplaced here. It would be better to fit it in earlier, before mentioning the evolved traits (lines 86-88).

Line 123: Add P:C ratio the given values equate to (1:13.5)

Line 131: Delete "them".

Lines 209-217: It would very helpful and good practice to publish the R code for all analyses. Why was triglyceride level and development rate not modelled in glmers, too, since they're ratios, as is survival? Were glmer with binomial error checked for overdispersion? Were any assumptions checked, and if so, how?

Line 336: "thrifty" needs more explanation.

Line 339: Typo: 'Is'.

Figure2 1b,c, 2: Add SE or 95% CI.

Table S3: Check consistency of interaction symbol (: or x).

Author's Response to Decision Letter for (RSPB-2020-2158.R0)

See Appendix A.

RSPB-2020-2684.R0

Review form: Reviewer 2

Recommendation

Accept as is

Scientific importance: Is the manuscript an original and important contribution to its field?

Good

General interest: Is the paper of sufficient general interest?

Good

Quality of the paper: Is the overall quality of the paper suitable?

Excellent

Is the length of the paper justified?

Yes

Should the paper be seen by a specialist statistical reviewer?

No

Do you have any concerns about statistical analyses in this paper? If so, please specify them explicitly in your report.

No

It is a condition of publication that authors make their supporting data, code and materials available - either as supplementary material or hosted in an external repository. Please rate, if applicable, the supporting data on the following criteria.

Is it accessible?

Yes

Is it clear?

Yes

Is it adequate?

Yes

Do you have any ethical concerns with this paper?

No

Comments to the Author

All comments have been addressed in detail. I don't have any further comments to add.

Review form: Reviewer 3

Recommendation

Accept with minor revision (please list in comments)

Scientific importance: Is the manuscript an original and important contribution to its field?

Excellent

General interest: Is the paper of sufficient general interest?

Good

Quality of the paper: Is the overall quality of the paper suitable?

Excellent

Is the length of the paper justified?

Yes

Should the paper be seen by a specialist statistical reviewer?

No

Do you have any concerns about statistical analyses in this paper? If so, please specify them explicitly in your report.

No

It is a condition of publication that authors make their supporting data, code and materials available - either as supplementary material or hosted in an external repository. Please rate, if applicable, the supporting data on the following criteria.

Is it accessible?

No

Is it clear?

N/A

Is it adequate?

N/A

Do you have any ethical concerns with this paper?

No

Comments to the Author

This manuscript presents results from a long-term experimental evolution study using *Drosophila melanogaster*. The work explores whether evolution on a diet that is both imbalanced and poor in overall nutrient availability, would lead to post-ingestive adaptations in nutrient assimilation. Specifically the authors test two competing hypotheses, i) that under evolution on poor imbalanced diets, natural selection will favour increased assimilation of all nutrients within the diet, or ii) evolution on the poor imbalanced diet will lead to more selectivity in nutrient assimilation in favour of limiting essential nutrients (such as protein). The authors find that Selected flies, those evolved on the poor imbalanced diet, showed increased protein assimilation but decreased carbohydrate assimilation. This coupled with the Selected flies showing lower triglyceride storage led the authors to conclude that these mechanisms favoured enhanced protein assimilation, rather than improving total nutrient assimilation. Finally, through using supplemented diets, the authors demonstrate that only protein was limiting in the poor imbalanced diets, not sucrose. The authors key conclusion is that evolution of nutrient compensation mechanisms can be driven by changes in abundance of nutrients despite the overall nutrient ratio being constant.

The manuscript is well written and presents novel and interesting results with thoughtful and well considered conclusions. The authors have appropriately considered and sensibly responded to the comments and suggestions from the two previous reviewers. Overall, I think these results would be of wide interest and a valuable addition to the field. I picked up on a few very minor typos which I detail below:

Line 135: "...from a given population to lay eggs..." - delete 'to'.

Line 169: "at similarly imbalanced P:C ratio," - add 'a' before similarly (or alternatively make ratio plural)

Line 302: You start two sentences in a row with "thus". Maybe change the second thus to a similar alternative or remove entirely (I think it reads fine without it).

Line 317: "...development on the poor diet larvae of..." - comma needed after diet.

Line 356: "...homogenous diet should..." - diet should be plural or you need "a" before homogenous.

Below I also list a few suggested improvements for the supplementary materials. Again, these are very minor and are not essential, but I think they would improve the supporting information:

Page 1, first paragraph: The sentence beginning “However, to eliminate medium heterogeneity....” is long a difficult to unpack. I suggest rewriting.

Generally, the resolution of the figures in the supplement is low. I suggest the authors improve this for the finished supplementary file.

The text size on some of the x and y-axis labels is quite small and should be increased (particularly figures S2 and S3).

Review form: Reviewer 4

Recommendation

Accept as is

Scientific importance: Is the manuscript an original and important contribution to its field?

Good

General interest: Is the paper of sufficient general interest?

Good

Quality of the paper: Is the overall quality of the paper suitable?

Good

Is the length of the paper justified?

Yes

Should the paper be seen by a specialist statistical reviewer?

No

Do you have any concerns about statistical analyses in this paper? If so, please specify them explicitly in your report.

No

It is a condition of publication that authors make their supporting data, code and materials available - either as supplementary material or hosted in an external repository. Please rate, if applicable, the supporting data on the following criteria.

Is it accessible?

No

Is it clear?

N/A

Is it adequate?

N/A

Do you have any ethical concerns with this paper?

No

Comments to the Author

I was not part of the original review of this manuscript but I have read the revised version and the original reviews along with the author responses. I enjoyed reading the revised manuscript and I think the authors have done a very good job responding to the concerns. I also have no problem with the speculation in the Discussion on how the requirements for different nutrients, or might depend on the total caloric/macronutrient content. I found the interpretation of the results to be balanced and the Discussion was particularly well written. The only issue I had was a bit of a struggle with 3rd paragraph of the Introduction. Attention is focused on the situation in which the consequences of diet imbalance are more severe when nutrients are scarce, with the alternative to selection for increased efficiency in assimilation of all nutrients being stronger selection for regulation on poorer diets (l 69-70). But it seems entirely plausible that selection on compensation could differ in rich vs. poor nutrient diets without necessarily being stronger in the poor (just different). Perhaps in rich diets C is more strongly linked to fitness, while in a poor diet N is. Selection will then favor a shift in compensation without necessarily being stronger in the poor diet. In other words, the alternative to the 'enhanced assimilation hypothesis' is broader than discussed, and what is presented is one of a larger family of possibilities.

Typo l 191 - extra 'of'

Line 117-118 - isn't (ii) a particular form of (i), making it confusing to word it this way? What about "...mediated by a shift in post-ingestive nutritional compensation arising from increased assimilation efficiency of the limiting nutrient."

Fig. S3. Square vs. triangle symbols are very hard to see. Perhaps dashed vs. solid lines would be preferable.

Decision letter (RSPB-2020-2684.R0)

10-Nov-2020

Dear Dr Kawecki

I am pleased to inform you that your manuscript RSPB-2020-2684 entitled "Experimental evolution of post-ingestive nutritional compensation in response to a nutrient-poor diet" has been accepted for publication in Proceedings B.

The referees have recommended publication, but also suggest some minor revisions to your manuscript. Therefore, I invite you to respond to the referees' comments and revise your manuscript. Because the schedule for publication is very tight, it is a condition of publication that you submit the revised version of your manuscript within 7 days. If you do not think you will be able to meet this date please let us know.

Sincerely,
 Professor Hans Heesterbeek
 mailto: proceedingsb@royalsociety.org

Associate Editor
 Board Member
 Comments to Author:

This new version of the paper has been seen by three reviewers with strong expertise in ageing, diet and the interaction between the two. All three experts are similarly happy with the paper and recommend acceptance. I don't think I have ever seen three reviewers simultaneously recommending acceptance in eight years as AE in this journal. Congratulations on a very nice paper that will make a fine contribution to the literature on this subject.

Reviewer(s)' Comments to Author:

Referee: 2

Comments to the Author(s).
 All comments have been addressed in detail. I don't have any further comments to add.

Referee: 3

Comments to the Author(s).
 This manuscript presents results from a long-term experimental evolution study using *Drosophila melanogaster*. The work explores whether evolution on a diet that is both imbalanced and poor in overall nutrient availability, would lead to post-ingestive adaptations in nutrient assimilation. Specifically the authors test two competing hypotheses, i) that under evolution on poor imbalanced diets, natural selection will favour increased assimilation of all nutrients within the diet, or ii) evolution on the poor imbalanced diet will lead to more selectivity in nutrient assimilation in favour of limiting essential nutrients (such as protein). The authors find that Selected flies, those evolved on the poor imbalanced diet, showed increased protein assimilation but decreased carbohydrate assimilation. This coupled with the Selected flies showing lower triglyceride storage led the authors to conclude that these mechanisms favoured enhanced protein assimilation, rather than improving total nutrient assimilation. Finally, through using supplemented diets, the authors demonstrate that only protein was limiting in the poor imbalanced diets, not sucrose. The authors key conclusion is that evolution of nutrient compensation mechanisms can be driven by changes in abundance of nutrients despite the overall nutrient ratio being constant.

The manuscript is well written and presents novel and interesting results with thoughtful and well considered conclusions. The authors have appropriately considered and sensibly responded to the comments and suggestions from the two previous reviewers. Overall, I think these results would be of wide interest and a valuable addition to the field. I picked up on a few very minor typos which I detail below:

Line 135: “..from a given population to lay eggs...” - delete 'to'.

Line 169: “at similarly imbalanced P:C ratio,” – add 'a' before similarly (or alternatively make ratio plural)

Line 302: You start two sentences in a row with “thus”. Maybe change the second thus to a similar alternative or remove entirely (I think it reads fine without it).

Line 317: "...development on the poor diet larvae of..." - comma needed after diet.

Line 356: "...homogenous diet should..." - diet should be plural or you need "a" before homogenous.

Below I also list a few suggested improvements for the supplementary materials. Again, these are very minor and are not essential, but I think they would improve the supporting information:

Page 1, first paragraph: The sentence beginning "However, to eliminate medium heterogeneity...." is long a difficult to unpack. I suggest rewriting.

Generally, the resolution of the figures in the supplement is low. I suggest the authors improve this for the finished supplementary file.

The text size on some of the x and y-axis labels is quite small and should be increased (particularly figures S2 and S3).

Referee: 4

Comments to the Author(s).

I was not part of the original review of this manuscript but I have read the revised version and the original reviews along with the author responses. I enjoyed reading the revised manuscript and I think the authors have done a very good job responding to the concerns. I also have no problem with the speculation in the Discussion on how the requirements for different nutrients, or might depend on the total caloric/macronutrient content. I found the interpretation of the results to be balanced and the Discussion was particularly well written. The only issue I had was a bit of a struggle with 3rd paragraph of the Introduction. Attention is focused on the situation in which the consequences of diet imbalance are more severe when nutrients are scarce, with the alternative to selection for increased efficiency in assimilation of all nutrients being stronger selection for regulation on poorer diets (l 69-70). But it seems entirely plausible that selection on compensation could differ in rich vs. poor nutrient diets without necessarily being stronger in the poor (just different). Perhaps in rich diets C is more strongly linked to fitness, while in a poor diet N is. Selection will then favor a shift in compensation without necessarily being stronger in the poor diet. In other words, the alternative to the 'enhanced assimilation hypothesis' is broader than discussed, and what is presented is one of a larger family of possibilities.

Typo l 191 - extra 'of'

Line 117-118 - isn't (ii) a particular form of (i), making it confusing to word it this way? What about "...mediated by a shift in post-ingestive nutritional compensation arising from increased assimilation efficiency of the limiting nutrient."

Fig. S3. Square vs. triangle symbols are very hard to see. Perhaps dashed vs. solid lines would be preferable.

Author's Response to Decision Letter for (RSPB-2020-2684.R0)

See Appendix B.

Decision letter (RSPB-2020-2684.R1)

11-Nov-2020

Dear Dr Kawecki

I am pleased to inform you that your manuscript entitled "Experimental evolution of post-ingestive nutritional compensation in response to a nutrient-poor diet" has been accepted for publication in Proceedings B.

Open Access

Paper charges

Sincerely,

Appendix A

Response to reviewers' comments (in blue)

Associate Editor

Board Member: 1

Comments to Author:

I think this is an interesting study that has a potential to contribute towards literature in the field of evolutionary responses to nutrition.

Two experts provided what I see as generally positive evaluation of this work. However, they also made suggestions towards improvement and expressed some concerns that need to be addressed. I suggest that the authors should pay close attention to the comments. I am particularly interested in their response to comment by Reviewer 1 regarding population differences in what constitutes optimal diet and how this may affect the interpretation of the data. Furthermore, the same reviewer makes important arguments regarding the interpretation of the results in the light of what we learned so far from nutritional geometry studies in *Drosophila*.

RESPONSE: We thank the reviewers and the editor for their effort and their generally positive evaluation of our ms.

We tried to do our best to address these comments. In one case, we were not sure what reviewer 1 meant exactly. We explained our position in the response below and will be willing to consider a more precise suggestion.

The data have been uploaded to Dryad and are accessible under https://datadryad.org/stash/share/PI1_6yLD45OYTrnKjjFmMffMJlbDhwoNp5SfTEzVJ84

Reviewer(s)' Comments to Author:

Referee: 1

Comments to the Author(s)

This manuscript makes use of a long-term selection experiment in fruit flies to understand how nutrient assimilation changes with adaptation to poor diet. The authors show that the selected flies increase their nitrogen assimilation and decrease their carbohydrate assimilation, and further that when reared on poor diet the selected lines store less triglycerides. Finally, they demonstrate that yeast, the primary protein source, is limiting for most larval traits in both the control and selected lines reared on poor diets. I think this manuscript presents novel and important findings that will be relevant to many in the field, however I have some suggestions that I feel should be addressed before publication.

The arguments around the optimal conditions for larval traits are not based on *Drosophila* populations used for these experiments. Recent studies have shown that populations differ in the way they respond

to quantitative variation in nutrition. This makes it difficult to say for certain that for your lab population the diets are imbalanced or off target. I think this needs to be acknowledged.

RESPONSE: We agree that the optimal diet for larval development may vary between populations of the same species, and thus that the optimal P:C ratio for our populations might deviate somewhat from that observed in Rodriguez et al. However, the optimal P:C ratio found by Rodriguez et al (between 1:2 and 1.5:1, depending of the focal performance trait) differs from the P:C ratio of our diet (1:13.5) by an order of magnitude. It is extremely unlikely that this would be an optimal diet for our populations, as confirmed by the effect of supplementing yeast (and no effect of supplementing sugar). We now make this point in the paper (l. 86-90) as follows: "Even allowing that the optimal diet of our populations may differ somewhat from that reported in [29], our poor diet is clearly protein-deficient and highly imbalanced."

Lines 124-126: I'm not sure I understand the selected diet. Here you state that the Selected lines were maintained on a diet that contained $\frac{1}{4}$ of the yeast, sugar, and cornmeal from the control diet. Because you've reduced all ingredients by the same amount you haven't actually changed the P:C ratio from the control diet right? So you are comparing between a rich imbalanced and poor imbalanced diet?

RESPONSE: Exactly; while we stated this elsewhere, this information was indeed missing where the diets were described in the Methods. We now explicitly state here (l. 131-132) that " the poor diet had the same 1:13.5 P:C ratio as the standard diet."

Given that most nutritional geometry experiments compare across P:C ratios (for the same caloric value), I think you need to make this very clear in the introduction that you are looking at selection to a total macronutrient dilution. Because larval traits are driven to a large extent by the protein concentration (and not the interactions between protein and carbohydrate) in their diets, I think this is still reasonable. However, you could also frame this as larvae adapt to low calorie diets by increasing nitrogen assimilation and decreasing carbohydrate assimilation. In fact, this argument would speak directly to one of the central themes in nutritional geometry: that many of the effects attributed to caloric dilution actually are a product of macronutrient concentration and have nothing to do with calories.

RESPONSE: Indeed, the main point we are making in our paper is that different diets may favor different degree of post-ingestive compensation even if both diets are equally unbalanced (have the same P:C ratio). The reviewer's comment made us realize we did not make this sufficiently clear. To amend this, we now explicitly state in the 3rd paragraph of discussion: "In this paper we use experimental evolution to study adaptation to such an imbalanced and nutrient-poor diet. Specifically, we ask if and how overall reduction of nutrient content affects the evolution of nutrient assimilation if the ratio of different nutrients remains constant." The rest of that paragraph discusses alternative hypotheses as to the populations may respond, reminding the reader on four occasions that we are talking about a situation when the nutrient ratios remain constant.

We have also modified the text of the following paragraph, where we introduce our experimental system, to drive home the point that the diets we use do not differ in the P:C ratio.

Lines 241-243: Why were the lines not compared on the rich diet as well? Would you expect them to differ? I think it's important to include in this portion of the results that triglyceride levels were measured in prepupae that had been raised on the poor diet. I realize that this is stated in the methods, but the reader doesn't always remember these details.

RESPONSE: In this paper we focus on evolutionary adaptation to the poor imbalanced larval diet, specifically on whether it involved changes in assimilation rate of the scarce and imbalanced nutrients. Addressing this question asks for a comparison of assimilation rates and TGA content in adapted (=Selected) and non-adapted (=Control) larvae on the poor diet. We now state this clearly both in the Results and in the Discussion.

We agree that it would be of interest to understand how adaptation to poor diet affects performance on the standard diet, e.g., if there are trade-offs. However, this would go beyond the scope of the paper, which is already at the upper length limit for the journal. We did in fact recently measure triglycerides in prepupae raised on standard diet, and found that the difference between Control and Selected populations is much less pronounced than on the poor diet. However, we do not have data on assimilation rates from the standard diet (which would require another labor-intensive and costly experiment) so it would be difficult to interpret the triglyceride data. Therefore we did not add the triglyceride data from standard food to this paper, we plan to publish them in another paper that would focus on changes in metabolism (an ongoing project).

Lines 255-257: The effects of sugar supplementation aren't significant.

RESPONSE: We have rephrased this sentence to state "No effects of sugar supplement were detected; if anything, it tended to worsen egg-to-adult survival probability". The $p = 0.08$ and the CI that includes zero are reported in the same sentence, so there is no ambiguity. The main point we want to make with this is that there is no hint of a positive effect of sugar supplement.

In the discussion, I think the points about the effects of P:C ratios are hard to make, because these weren't varied in your experimental design (the poor and control diet are on the same nutrient rail). I think these parts should be toned down, and proposed as suggestions.

RESPONSE: We find it difficult to respond to this comment because of its vagueness. In the discussion we talk about P:C ratio in three different contexts: (1) the P:C ratio of the diet, (2) the P:C ratio in the nutrients assimilated into the organism, and (3) the putative optimal P:C ratio of nutrients. It is not clear to which of these the reviewer's criticism is directed.

For the P:C ratio in the diet, the reviewer is correct that we did not vary dietary P:C ratio as a treatment in our experiments, but we make no claims anywhere about effects of different dietary P:C ratios, except where summarizing published work (l. 356-365).

A major message of our paper is that differences in post-ingestive nutritional compensation may evolve even if the dietary P:C ratio remains identical. This is a novel conclusion; nutritional compensation seems to have so far been exclusively studied in the context of differences in dietary P:C ratio. This conclusion is based on a clear experimental result: we had two diets with the same P:C ratio, yet, the populations evolved a shift in the relative assimilation of amino-acids vs carbohydrates. Of course, how

general this is remains to be seen, but for this experimental system the evidence allows us to present it as a conclusion rather than as a mere suggestion.

We also talk in the Discussion about the P:C ratio of the assimilated nutrients as an evolutionary response to poor vs standard diet. Our data clearly indicate that a shift towards a higher P:C in assimilated nutrients in the Selected populations compared to Control populations. Thus, again we feel we are justified to make this conclusion.

Finally, the Discussion also includes speculations as to why the Selected and Control populations show a different degree of dietary compensation despite the two diets having the same P:C ratio. These speculations invoke in part the putative optimal P:C ratio (l. 373-399). These are indeed speculations and we do not have data that would support or refute them directly. They do not pertain to the effects of changing dietary P:C ratios, but rather how the requirements of the organism for different nutrients, or consequences of suboptimal nutrient ratios, might depend on the total calorie/macronutrient content of the diet. We think a limited amount of speculation is permissible in a Discussion, and note that Reviewer 2 actually asked for such putative explanations. Therefore, we retained them for the moment. However, we can remove them if required.

Minor comments:

Lines 44-45, 61, 364-365: The geometric framework for nutrition isn't really a theory. Rather, it is a framework that allows you to describe how traits respond to quantitative variation in nutrients. This framework has led to several hypotheses, like the protein leveraging hypothesis, but doesn't postulate any particular response to diet per se. Initially, this framework was devised to understand how animals make foraging decisions. These studies found that animals often make foraging decisions that are based around achieve an intake target of one or a combination of nutrients. More recently it has been adopted to understand how life history traits respond to quantitative variation in nutrient conditions. It's important to note that, while uncommon, intake targets can theoretically revolve around maintaining constant calorie concentrations.

RESPONSE: This is a good point, and indeed most of our alternative hypotheses can be accommodated in this framework. We thus changed "theory" to "framework" and generally removed statements that attributed specific predictions as emerging from a "geometric theory". Still, at least based on our reading to several review/conceptual papers on the framework, two empirically testable notions/hypotheses are inexorably linked to this framework: (1) that ratios of nutrients are highly important and not just the total caloric content of the diet, and (2) that imbalance diet should favor nutritional compensation. Thus we feel justified to retain linking these notions to the geometric framework (l. 12-13, l. 44-46).

Referee: 2

Comments to the Author(s)

In this manuscript, the authors describe an experimental evolution study in the fly *Drosophila melanogaster* to test for the evolution of post-ingestive nutritional compensation as an adaptation to

lowered total dietary nutrient content at the larval stage.

After between 233 and 256 generations on either ancestral (n=6), or low-nutrient diet (n=6), with imbalanced P:C ratio of 1:13 for all diets, flies evolved enhanced assimilation of nitrogen from dietary yeast and reduced assimilation of carbon from sucrose. Responses were measured using analyses of stable nitrogen and carbon isotopes, triglyceride levels, and survival and development rate.

I would have wished to read about the possible mechanisms of the observed evolutionary changes. If this is based mainly on standing genetic variation that existed in the stock lab population at the start of the experiment, is there a reason why a more efficient uptake of nitrogen wouldn't evolve, even under much lower selection, i.e. is there an assumed trade-off?

RESPONSE: We have devoted the 4th paragraph of Discussion to the potential mechanisms of the observed evolutionary changes in assimilation of nitrogen versus carbon and reduced accumulation of triglycerides. Six potential mechanisms are discussed there (selective feeding, changes in feeding rate, shifts in the production of digestive enzymes, efficiency of transport across the intestinal wall, increased catabolism of carbohydrates, lower catabolism of amino-acids).

Concerning the genetic variation underlying the response, given the time scale of the experiment and the population sizes, the evolutionary change must have been essentially entirely based on standing genetic variation (doi:10.1534/genetics.104.036947). We have now added a statement to this effect and the reference (l. 368-370).

Concerning potential role of trade-offs, indeed, as the reviewer suggests, weaker selection for compensating mechanisms might not be sufficient to overcome their costs. We now state this explicitly (l. 376).

Minor comments:

Line 90: Referring to the control diet' nutrient concentration, "thus, it has the same imbalanced P:C ratio": I don't see how 'thus' makes sense here, linking to the previous statement about nutrient concentration.

RESPONSE: the fact that the P:C ratio of the two diets is the same (the second clause) is a consequence of the fact that the amount of all nutrients was changed by the same factor (the first clause). If different nutrients were changed by different factors, the P:C would not remain constant. So "thus" is justified, and we retained it.

Also, the description of the control diet feels a bit misplaced here. It would be better to fit it in earlier, before mentioning the evolved traits (lines 86-88).

RESPONSE: We rearranged the text as suggested by the reviewer, so that the control diet is now introduced before we talk about how the Selected populations evolved.

Line 123: Add P:C ratio the given values equate to (1:13.5)

RESPONSE: Done

Line 131: Delete "them".

RESPONSE: Done

Lines 209-217: It would very helpful and good practice to publish the R code for all analyses. Why was triglyceride level and development rate not modelled in glmers, too, since they're ratios, as is survival? Were glmer with binomial error checked for overdispersion? Were any assumptions checked, and if so, how?

RESPONSE: We used the GLMM for survival because the data for this trait consist of integer counts of k successes (= survivors) out of n trials (= initial number of eggs put in the vial); the sampling error in this case is expected to follow a binomial distribution. So this justifies the use of GLMM with binomial distribution and logit link. Concerning overdispersion, the model included a random effect of the rearing bottle (which would be a main source of overdispersion). Nonetheless, we did check for overdispersion using "overdisp" function of the sjstats package; all ratios were <1.5 (all $P > 0.2$); hence, we did not include an overdispersion parameter in the model. We have added this information to the Supplementary Methods.

Developmental rate is the inverse of developmental time, and thus a continuous variable. While there are several other continuous distributions implemented in lme4, inspection of residuals from LMM indicated that they conform to a normal distribution. Triglyceride content has still a different nature, being a ratio of two independently taken continuous measurements; as such, we use its log in the analysis. Again, we have no a priori reasons why the distribution of this variable should not be normal, and the residuals from the LMM indeed conform to normal distribution. So we retained LMM for these variables (which in fact is equivalent to GLMM with normal distribution and identity link). We added the information about normality test to the Supplementary Methods.

Concerning R scripts, we are not entirely convinced that including them in supplementary material is useful, given that our analysis are just standard LMMs and GLMMs, but we have now added a separate supplementary file with the parts of the script that concern the model fitting, significance tests and marginal means estimation.

While revising the paper, we realized that we had been inconsistent in how we treated experimental blocks (random in the stable isotope experiment, fixed in the diet supplement experiment). To be consistent, we redid the analysis of the former experiment with block as fixed (because there are only 3 levels, it is better to model it as fixed, even though it is in principle random). This resulted in small changes in F and P , with no changes to the conclusions.

Line 336: "thrifty" needs more explanation.

We mean to say that the Selected larvae seem thrifty with their energy expenditure, rather than burning off extra carbohydrates to generate more ATP. We have rephrased this part of the text.

Line 339: Typo: 'ls'.

RESPONSE: corrected

Figure2 1b,c, 2: Add SE or 95% CI.

RESPONSE: We agree that this is important information. However, we prefer to plot original data points in the figures, and adding SE bars on top would overload the figure, making it hard to read. Therefore, we have now included the means and SE for the two sets of populations in the text of the Results section.

Table S3: Check consistency of interaction symbol (: or x).

RESPONSE: corrected (we now use "x"). (However, where we quote the R script we use the ":" notation.)

Appendix B

Dear Editors,

here are our responses to the minor comments by the reviewers, followed by the ms file indicating the changes in red.

Thank you for your input that helped us improve the paper – and of course for the favorable decision.

Sincerely

Tadeusz J. Kawecki

Associate Editor

Board Member

Comments to Author:

This new version of the paper has been seen by three reviewers with strong expertise in ageing, diet and the interaction between the two. All three experts are similarly happy with the paper and recommend acceptance. I don't think I have ever seen three reviewers simultaneously recommending acceptance in eight years as AE in this journal.

Congratulations on a very nice paper that will make a fine contribution to the literature on this subject.

RESPONSE: We are pleased that our efforts were appreciated.

Reviewer(s)' Comments to Author:

Referee: 2

Comments to the Author(s).

All comments have been addressed in detail. I don't have any further comments to add.

Referee: 3

Comments to the Author(s).

This manuscript presents results from a long-term experimental evolution study using *Drosophila melanogaster*. The work explores whether evolution on a diet that is both imbalanced and poor in overall nutrient availability, would lead to post-ingestive adaptations in nutrient assimilation. Specifically the authors test two competing hypotheses, i) that under evolution on poor imbalanced diets, natural selection will favour increased assimilation of all nutrients within the diet, or ii) evolution on the poor imbalanced diet will lead to more selectivity in nutrient assimilation in favour of limiting essential nutrients (such as protein). The authors find that Selected flies, those evolved on the poor imbalanced diet, showed increased protein assimilation but decreased carbohydrate assimilation. This coupled with the Selected flies showing lower triglyceride storage led the authors to conclude that these mechanisms favoured enhanced protein assimilation, rather than improving total nutrient assimilation. Finally, through using supplemented diets, the authors demonstrate that only protein was limiting in the poor imbalanced diets, not sucrose. The authors key conclusion is that evolution of nutrient compensation mechanisms can be driven by changes in abundance of nutrients despite the overall nutrient ratio being constant.

The manuscript is well written and presents novel and interesting results with thoughtful and well considered conclusions. The authors have appropriately considered and sensibly responded to the comments and suggestions from the two previous reviewers. Overall, I think these results would be of wide interest and a valuable addition to the field. I picked up on a few very minor typos which I detail below:

Line 135: “..from a given population to lay eggs...”- delete 'to'.

DONE

Line 169: “at similarly imbalanced P:C ratio,” – add 'a' before similarly (or alternatively make ratio plural)

DONE

Line 302: You start two sentences in a row with “thus”. Maybe change the second thus to a similar alternative or remove entirely (I think it reads fine without it).

RESPONSE: We removed the second “thus”.

Line 317: “...development on the poor diet larvae of...” – comma needed after diet.

DONE

Line 356: “...homogenous diet should...” – diet should be plural or you need “a” before homogenous.

RESPONSE: we changed to plural

Below I also list a few suggested improvements for the supplementary materials. Again, these are very minor and are not essential, but I think they would improve the supporting information:

Page 1, first paragraph: The sentence beginning “However, to eliminate medium heterogeneity....” is long a difficult to unpack. I suggest rewriting.

RESPONSE: We have elaborated on this as follows: " However, in the nutrient assimilation experiment we used a medium without cornmeal, with yeast and sucrose as the only sources of nutrients. We did so because cornmeal grains, being relatively coarse, create a degree of heterogeneity in the medium that might allow larvae to prefer or avoid them selectively, and our interest was in postingestive rather than preingestive compensation. The absence of cornmeal also made it easier to remove the larvae from the medium. While often used, cornmeal is not a necessary ingredient of *Drosophila* diet."

Generally, the resolution of the figures in the supplement is low. I suggest the authors improve this for the finished supplementary file.

The text size on some of the x and y-axis labels is quite small and should be increased (particularly figures S2 and S3).

RESPONSE: We polished the supplementary figures and made sure they are sufficient quality (300 dpi) in the final version.

Referee: 4

Comments to the Author(s).

I was not part of the original review of this manuscript but I have read the revised version and the original reviews along with the author responses. I enjoyed reading the revised manuscript and I think the authors have done a very good job responding to the concerns. I also have no problem with the speculation in the Discussion on how the requirements for different nutrients, or might depend on the total caloric/macronutrient content. I found the interpretation of the results to be balanced and the Discussion was particularly well written. The only issue I had was a bit of a struggle with 3rd paragraph of the Introduction. Attention

is focused on the situation in which the consequences of diet imbalance are more severe when nutrients are scarce, with the alternative to selection for increased efficiency in assimilation of all nutrients being stronger selection for regulation on poorer diets (l. 69-70). But it seems entirely plausible that selection on compensation could differ in rich vs. poor nutrient diets without necessarily being stronger in the poor (just different). Perhaps in rich diets C is more strongly linked to fitness, while in a poor diet N is. Selection will then favor a shift in compensation without necessarily being stronger in the poor diet. In other words, the alternative to the 'enhanced assimilation hypothesis' is broader than discussed, and what is presented is one of a larger family of possibilities.

RESPONSE: We modified the text slightly to encompass this more general possibility (added text in red):

" Alternatively, even if the nutrient ratios are the same, selection for regulation and compensation might be affected by the overall caloric content of the diet. In particular, a non-optimal ratio may not be too detrimental if there is a lot of nutrients and become more critical when the diet is generally poor, resulting in a stronger selection on compensation."

We also discuss the possibility of different optima for P:C on poor versus rich diets in the Discussion (l. 378-383).

Typo l 191 - extra 'of'

CORRECTED

Line 117-118 - isn't (ii) a particular form of (i), making it confusing to word it this way? What about "...mediated by a shift in post-ingestive nutritional compensation arising from increased assimilation efficiency of the limiting nutrient."

RESPONSE: The reviewer is right that (ii) is a special case of (i); however, focusing on the increased assimilation of the limiting nutrient (protein) is only half of the story of our paper, the shift is also mediated by lower assimilation of sugars. Thus we edited this part to "mediated by a shift in post-ingestive nutritional compensation arising from both a higher assimilation efficiency of the limiting nutrient and a lower assimilation rate of a non-limiting nutrient."

Fig. S3. Square vs. triangle symbols are very hard to see. Perhaps dashed vs. solid lines would be preferable.

RESPONSE: The two symbols indicated different replicates, and there is no specific relationship between replicate 1 for population or treatment X and replicate 1 for population or treatment Y. Thus, we decided to use the same symbols for the two replicates to simplify the plots.